# General-Reasoner: Advancing LLM Reasoning Across All Domains

♥*Xueguang Ma, ♦*Qian Liu, ♥♣Dongfu Jiang, ♥♣Ge Zhang, ♦Zejun Ma, ♥♣Wenhu Chen

♥University of Waterloo, ♦Vector Institute, ♦TikTok, Singapore, ♣M-A-P
x93ma@uwaterloo.ca,wenhuchen@uwaterloo.ca

https://tiger-ai-lab.github.io/General-Reasoner/

## Abstract

Reinforcement learning (RL) has recently demonstrated strong potential in enhancing the reasoning capabilities of large language models (LLMs). Particularly, the "Zero" reinforcement learning introduced by Deepseek-R1-Zero, enables direct RL training of base LLMs without relying on an intermediate supervised fine-tuning stage. Despite these advancements, current works for LLM reasoning mainly focus on mathematical and coding domains, largely due to data abundance and the ease of answer verification. This limits the applicability and generalization of such models to broader domains, where questions often have diverse answer representations, and data is more scarce. In this paper, we propose GENERAL-REASONER, a novel training framework designed to enhance LLM reasoning capabilities across diverse domains. Our key contributions include: (1) constructing a large-scale, high-quality dataset of questions with verifiable answers curated by web crawling, covering a wide range of disciplines; and (2) developing a generative model-based answer verifier, which replaces traditional rule-based verification with the capability of chain-of-thought and context-awareness. We train a series of models and evaluate them on a wide range of datasets covering wide domains like physics, chemistry, finance, electronics etc. Our comprehensive evaluation across these 12 benchmarks (e.g. MMLU-Pro, GPQA, SuperGPQA, TheoremQA, BBEH and MATH AMC) demonstrates that GENERAL-REASONER outperforms existing baseline methods, achieving robust and generalizable reasoning performance while maintaining superior effectiveness in mathematical reasoning tasks.

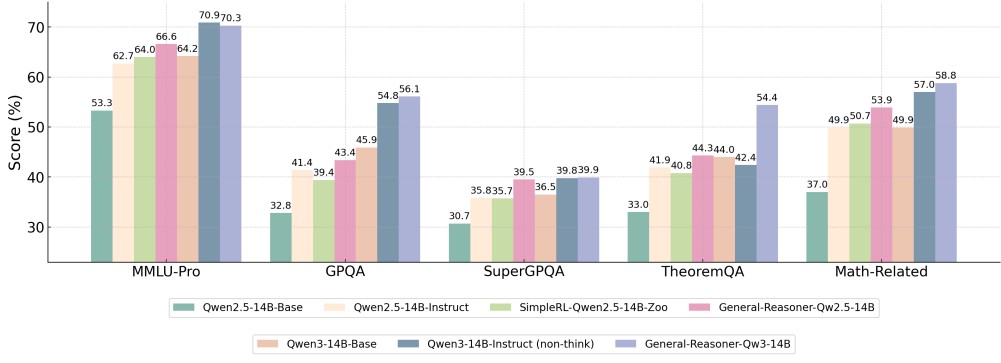

Figure 1: Effectiveness of our GENERAL-REASONER trained with diverse verifiable reasoning questions using model-based verifier compared to baseline methods on various reasoning tasks.

---

*The first two authors have equal contribution.

39th Conference on Neural Information Processing Systems (NeurIPS 2025).

# 1 Introduction

Recent advancements in large language models (LLMs) have demonstrated strong potential in many expert-level tasks by following user instructions. One of the most significant breakthroughs in recent work is the use of reinforcement learning (RL), such as PPO and GRPO, to improve the reasoning capabilities of LLMs. A particularly noteworthy development is the "Zero" reinforcement learning setting proposed by DeepSeek-R1-Zero [1], which shows that training a base LLM directly via RL can unlock powerful reasoning capabilities without relying on a supervised fine-tuning step. RL for LLM reasoning has inspired multiple open-source efforts, such as SimpleRL [2], DAPO [3], and DeepScaleR [4, 5], which leverage variants of GRPO [6] to further enhance reasoning performance, either in the zero RL setting or via continuous reinforcement learning on models with supervised fine-tuning or distillation.

However, existing methods primarily focus on training and evaluation in mathematical reasoning or coding tasks. This narrow focus can be attributed to two main reasons: (1) DATA ABUNDANCE: it is easier to harvest large-scale mathematical data from the Internet due to the abundance of international math and coding competitions and exams; (2) ANSWER VERIFICATION: mathematical domains allow for easy and reliable answer verification using straightforward rule-based methods (e.g., exact numeric matching, symbolic equation comparison), which provide accurate reward signals. However, this reliance on rule-based verifiers restricts the generalizability of the resulting models. Real-world reasoning often spans multiple disciplines, such as science, finance, and the humanities, and involves complex, long-tailed answer representations that cannot be reliably verified through heuristic rules. In fact, training solely on mathematical or coding data does not guarantee generalization to other reasoning domains. For example, although S1/S1.1 [7] significantly improves mathematical scores, it degrades performance on MMLU-Pro [8] by 4–6%.

To address these challenges and broaden the applicability and robustness of RL-based reasoning models, we propose training framework designed to enhance the reasoning capabilities of LLMs across diverse, non-mathematical domains, while preserving their strengths in mathematics. Our approach tackles the two aforementioned problems:

- **All-Domain Reasoning Dataset (WebInstruct-verified):** A major bottleneck for scaling reasoning beyond mathematics is the lack of publicly available, high-quality reasoning datasets that span multiple domains and offer reliably verifiable answers. To address this, we construct a large-scale, diverse-domain dataset by carefully crawling and filtering high-quality reasoning questions from web resources, based on WebInstruct [9]. We then employ state-of-the-art LLMs to automatically select questions whose answers can be reliably verified, significantly expanding the training scope to disciplines such as physics, chemistry, social sciences, and finance. Our carefully curated dataset contains approximately 230K high-quality reasoning questions, providing a robust foundation for general reasoning training across multiple complex domains.
- **All-Domain Answer Verifier (General-Verifier):** In addition to scaling data, existing rule-based verifiers struggle with the diverse answer types commonly found in broader domains, including short string answers, LaTeX expressions, and other structured forms, limiting the effectiveness of reasoning training beyond mathematics. We introduce a compact generative verifier model (1.5B parameters) explicitly trained to verify short answers in a context-aware, chain-of-thought manner, thereby providing robust and reliable reward signals for RL training. This approach improves the flexibility and scalability of RL training across diverse reasoning tasks.

Using this diverse, verifiable reasoning data along with the generative model-based verifier, we are able to train a series of GENERAL-REASONER models in the Zero-RL setting from various base models. To validate the effectiveness of our approach, we conduct comprehensive evaluations across 12 challenging reasoning benchmarks beyond mathematics, including MMLU-Pro [8], GPQA [10], SuperGPQA [11], TheoremQA [12], and BBEH [13], as well as standard mathematical reasoning benchmarks such as MATH-500 [14], GSM8K [15], and Olympiad [16]. GENERAL-REASONER typically boosts performance on general benchmarks like MMLU-Pro and SuperGPQA by approximately 10%. On math benchmarks, GENERAL-REASONER can even slightly outperform math-focused RL frameworks such as SimpleRL [2], benefiting from cross-domain generalization. Our best model GENERAL-REASONER-Qw3-14B is able to match or beat GPT-4o in various benchmarks.

In summary, our work makes the following key contributions:

1. We construct and release a large-scale, high-quality dataset (WebInstruct-verified) of verifiable reasoning questions with short-form answers spanning diverse domains, significantly broadening the training resources available for general reasoning.
2. We introduce a compact, generative model-based verifier (General-Verifier) specifically trained for chain-of-thought answer verification, effectively replacing rule-based methods and enabling robust and scalable RL training across multiple complex reasoning domains.
3. Through extensive evaluations, we empirically demonstrate the superior performance and generalization capabilities of GENERAL-REASONER in the Zero-RL setting, providing a strong baseline and valuable insights for future advancements in RL-driven general reasoning.

## 2 Related Works

### 2.1 Large Language Models for Reasoning

Advancements in large language models (LLMs) have demonstrated substantial potential in effectively performing diverse tasks by following user instructions. However, expert-level tasks, such as solving complex mathematical problems [12] or addressing practical STEM challenges [8], require robust reasoning capabilities. Recent work on chain-of-thought reasoning has shown that LLMs can significantly enhance their performance by explicitly decoding their thought processes during inference [17]. Subsequent studies further investigate the scaling of reasoning process, termed "test-time scaling," to better enhance the reasoning capability of LLMs [18–20].

Following this direction, commercial models such as OpenAI's O1 series have exhibited impressive performance on reasoning-intensive evaluations [21], including mathematical Olympiad problems. Meanwhile, recent open-source models, such as Qwen [22], QWQ [23], and Deepseek R1 [1], have also achieved competitive results, narrowing the performance gap with state-of-the-art commercial counterparts. This progress opens new avenues for academia to explore advanced optimization strategies aimed at further improving the reasoning capabilities of LLMs [7, 3, 24].

### 2.2 Zero Reinforcement Learning for LLMs

Typical methods for enhancing the reasoning capabilities of LLMs usually involve first performing supervised fine-tuning with chain-of-thought (CoT) data on a base model [25], followed by reinforcement learning (RL) to further improve reasoning and generalization performance. However, recent studies exemplified by Deepseek-R1-Zero [4] have demonstrated that directly applying reinforcement learning to a strong base model can effectively uncover significant reasoning capabilities without initial supervised fine-tuning. This "Zero Reinforcement Learning" approach has attracted considerable attention within the research community, inspiring efforts to replicate and enhance the method on other powerful base models such as Qwen [26].

A key advantage of zero RL is its efficiency in collecting verifiable question–answer pairs, which eliminates the need for complete reasoning chains as training targets. This efficiency provides greater flexibility in enhancing reasoning performance without complex data collection efforts, enabling models to self-improve more effectively. However, existing works on zero RL have mostly focused on mathematical reasoning [2]. While there are concurrent works [27, 28] aim to expand reasoning capabilities to broader domains, our work seeks to do so by leveraging a model-based verifier to scale diverse reasoning data across domains and provide comprehensive evaluations to demonstrate effectiveness.

## 3 General Reasoner

### 3.1 Diverse Verifiable Reasoning Tasks

To facilitate robust reasoning capabilities across a wide range of domains beyond mathematical problems, we construct a large-scale, diverse, and high-quality dataset composed of verifiable reasoning tasks. Our dataset-building pipeline is illustrated in Figure 2.

The initial data source is from WebInstruct dataset [9], which consists of around 5 million naturally occurring, web-crawled instructions from high-quality resource websites, including platforms like

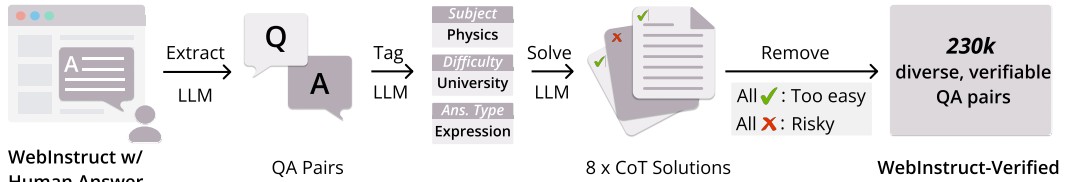

Figure 2: Data creation pipeline: It consists of QA mining, Extraction and Verification.

StackExchange and various educational portals. Despite WebInstruct's suitability for general instruction tuning, the majority of documents are not directly usable as reasoning tasks due to the lack of explicit verifiable answers or require a reasoning process.

To address this, we first trace the entries in WebInstruct back to its original web page to re-crawl precise question-answer pairs. During this process, we remove the questions lacking clearly identifiable human-written answers on the original source websites. Many websites require membership or complex interaction to show the answers, which will be filtered out. This careful selection aims to ensure retained entries are verified by humans, enhancing the dataset's reliability and correctness.

Next, we further leverage Gemini-1.5-Pro [29], a state-of-the-art LLM, to extract questions explicitly identified as having clearly verifiable short answers for single-turn questions. This step yields an intermediate dataset of approximately 1 million verifiable reasoning questions across various disciplines. Subsequently, we apply Gemini-2.0-Flash to annotate each question with metadata, including the answer type, subject category, and difficulty level. Recognizing the skewed ratio of mathematical tasks, we specifically filter out mathematics problems that are labeled as easier than university-level to ensure a more balanced and challenging dataset distribution.

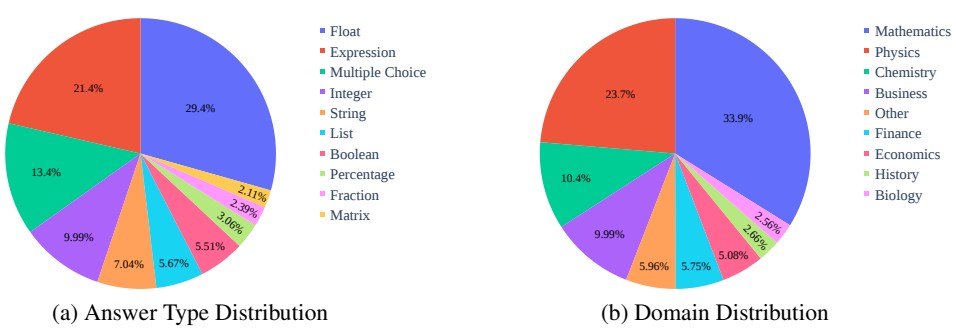

(a) Answer Type Distribution        (b) Domain Distribution

Additionally, recognizing that web-crawled data inherently contains noise, such as questions that are either unsolvable or trivially easy, we implement further rigorous filtering to refine the dataset quality. Specifically, Gemini-2.0-Flash generates eight candidate solutions for each question, allowing us to apply the following quality control criteria:

- We exclude questions for which all eight candidate solutions fail, effectively removing ambiguous or noisy questions that likely arise from crawling errors or incomplete source content.
- We also exclude overly simplistic questions for which all eight candidate solutions are correct, ensuring the dataset maintains sufficient complexity and presents meaningful challenges for robust reasoning and generalization during RL training.

The Gemini-2.0-flash generated solutions are also later utilized to train our proposed model-based verifier, which will be discussed in detail in the next section.

Eventually, the processed dataset contains approximately 230k reasoning questions spanning diverse answer formats, including multiple-choice, numerical expressions, matrices as highlighted in Figure 3a. Figure 3b further illustrates the balanced domain distribution of our curated dataset, encompassing disciplines such as mathematics, physics, chemistry, finance, and various other humanities and social sciences fields. This rigorous data curation process produces a challenging but reliable dataset for training generalizable reasoning capabilities in large language models.

Table 1: Examples of reasoning questions where the model provides correct answers, but the rule-based verifier fails to recognize their correctness, while the model-based verifier succeeds.

| | Example 1 | Example 2 | Example 3 |
|---|---|---|---|
| Question | Consider the line perpendicular to the surface $z = x^2 + y^2$ at the point where $x = 4$ and $y = 1$. Find a vector parametric equation for this line in terms of the parameter $t$. | Find the partial pressure in a solution containing ethanol and 1-propanol with a total vapor pressure of 56.3 torr. The pure vapor pressures are 100.0 torr and 37.6 torr, respectively, and the solution has a mole fraction of 0.300 of ethanol. | What is the work done to push a 1 kg box horizontally for 1 meter on a surface with a coefficient of friction of 0.5? |
| Ground Truth Answer | x = 4 + 8t, y = 1 + 2t, z = 17 - t | 30.0 torr, 26.3 torr | 4.9 J |
| Student Answer | 4 + 8t, 1 + 2t, 17 - t | The partial pressure of ethanol is 30.0 torr and the partial pressure of 1-propanol is 26.32 torr. | 4.9 N·m |
| Rule Based Verifier | False | False | False |
| Model Based Verifier | True | True | True |

## 3.2 Model-Based Verifier for GRPO

**Preliminary.** We adopt Group Relative Policy Optimization (GRPO) [6] following the recent advancements such as DeepSeek-R1 [1]. Given a question-answer pair $(q, a)$, a behavior policy $\pi_{\theta_{old}}$ samples a group of $G$ individual responses $\{o_i\}_{i=1}^{G}$. The GRPO objective updates model parameters $\theta$ as follows [3]:

$$\mathcal{J}_{\text{GRPO}}(\theta) = \mathbb{E}_{(q,a)\sim\mathcal{D}, \{o_i\}_{i=1}^{G}\sim\pi_{\theta_{old}}(\cdot|q)} \tag{1}$$

$$\left[ \frac{1}{G} \sum_{i=1}^{G} \frac{1}{|o_i|} \sum_{t=1}^{|o_i|} \left( \min\left( r_{i,t}(\theta)\hat{A}_{i,t}, \text{clip}(r_{i,t}(\theta), 1 - \epsilon, 1 + \epsilon)\hat{A}_{i,t} \right) - \beta D_{\text{KL}}(\pi_\theta \| \pi_{\text{ref}}) \right) \right],$$

where

$$r_{i,t}(\theta) = \frac{\pi_\theta(o_{i,t} \mid q, o_{i,<t})}{\pi_{\theta_{old}}(o_{i,t} \mid q, o_{i,<t})}, \qquad \hat{A}_{i,t} = \frac{r_i - \text{mean}(\{R_i\}_{i=1}^{G})}{\text{std}(\{R_i\}_{i=1}^{G})}. \tag{2}$$

In this work, our design on the model-based verifier specifically affects how the reward $R$ is computed.

**Limitations of Existing Reward Model.** Traditional reward models are trained through human feedback or preference assessment, returning scalar values based on the entire output to indicate overall quality. Although intuitive, these models are suffering from being hacked by the policy model, and usually require the reward model to have a large parameter size to be effective and robust.

In contrast, rule-based verifiers, widely used in mathematical reasoning due to simplicity, evaluate only the final answer, allowing models greater freedom to explore diverse reasoning paths. However, these rule-based approaches encounter several critical limitations when extending beyond mathematics:

- **Rigid Matching Criteria:** Rule-based methods typically require exact matches or adherence to rigid structures, failing to recognize semantically equivalent answers that differ in representation.
- **Semantic Insensitivity:** They are ineffective at interpreting answers that vary semantically, such as equivalent expressions or answers expressed in different units or formats.
- **Lack of Generality:** Adapting rule-based verification to a wide range of disciplines and diverse answer formats can be difficult, limiting their applicability and scalability.

**Generative Model-Based Verifier.** We introduce a compact generative model-based verifier specifically trained to robustly assess answer equivalence across diverse domains. Ideally, a state-of-the-art large language model (LLM) like Gemini-2.0 could verify answer equivalence; however, such solutions are computationally expensive and impractical for large-scale RL training.

Instead, we leverage our dataset creation pipeline, specifically Gemini-2.0-generated candidate answers and verification annotations, to train a compact 1.5B-parameter generative verifier model.

This verifier, initialized from Qwen2.5-Math-1.5B [26], is fine-tuned to assess student-generated short answers (extracted from the response) $s$ against ground-truth references $g$ in a generative manner, whose inference process is formulated as:

$$\hat{y} \sim P(y \mid q, g, s), \quad \hat{y} = y_{cot}, [sep], y_{label} \tag{3}$$

where $\hat{y}$ is a chain-of-thought reasoning process $y_{cot}$ with a final binary prediction $\hat{y}_{label}$ (true/false) on whether the student answer is equivalent to the ground-truth in the question context.

This verifier integrates seamlessly into our reinforcement learning pipeline, providing robust, accurate reward signals. Empirical analysis confirms that our model-based verifier achieves high agreement with Gemini-2.0-Flash, substantially outperforming traditional rule-based approaches.

## 4 Experiments

### 4.1 Training

We follow the Zero RL setting, directly conducting reinforcement learning (RL) from base large language models without an intermediate supervised fine-tuning stage. Specifically, we initialize our models using the base model from Qwen2.5 family (7B and 14B) and the newer Qwen3 family (4B and 14B) [22], and apply the GRPO algorithm. Please note that we pick the general Qwen model instead of the Qwen-math model to maximize the general performance, which could lead to seemingly lower performance on mathematical benchmarks. Reward scores during training are calculated as follows:

- If the solution extraction fails (e.g., no boxed answer or summarization such as "the solution is:"), the reward is -0.5.
- If the solution passes verification, the base reward is 1, with a length-based penalty applied to discourage excessively long generations. In our early experiments, we notice the model can hack the model-based verifier by putting long content in answer box if there is no length penalty.

```
penalty = -0.05 × min(10, abs(len_of_ground_truth - len_of_answer))
```

Training is conducted on 4 nodes with 8×H100 GPUs per node for up to 700 steps for Qwen2.5 series models. For model initialzied with Qwen3-Base, we train up to 400 steps. Please see Appendix A.5 for detailed hyperparameters of each model checkpoint. During training, the average model response length increases from approximately 700 tokens to around 1000 tokens. The total training time is around 2 days for the 4B/7B model and around 4 days for the 14B model. Our implementation is based on the `verl` repository.[2]

### 4.2 Evaluation

To evaluate the models' general reasoning capabilities, we conduct a comprehensive assessment across several challenging benchmarks:

- **MMLU-Pro** [8]: A robust and challenging massive multi-task understanding dataset tailored to more rigorously benchmark large language models' capabilities.
- **SuperGPQA** [11]: A large-scale benchmark targeting graduate-level reasoning across 285 diverse disciplines. All the questions are verified to be not found on Google Search.
- **BBEH** [13]: A new benchmark that extends BIG-Bench Hard [30] by introducing more challenging tasks for better evaluation of complex reasoning.
- **GPQA** [10]: Graduate-level question answering designed to be resistant to shallow pattern-matching or memorization. We use the diamond split in GPQA.
- **TheoremQA** [12]: Graduate-level question answering designed to require knowing corresponding theorems. It covers math, physics, EE&CS, and Finance. It's a general reasoning benchmark.
- **Math-Related Tasks**: A suite of standard math reasoning benchmarks, including MATH-500 [14], Olympiad [16], Minerva [31], GSM8K [15], AMC, AIME24, and AIME25. We use the `simple-evals`[3] evaluation framework, and use GPT4o [32] to check the answer equivalence.

---

[2] https://github.com/volcengine/verl

[3] https://github.com/openai/simple-evals

Table 2: Accuracy comparison of our GENERAL-REASONER with baseline methods on general reasoning benchmarks. MMLU-Pro, SuperGPQA, TheoremQA and BBEH contain multiple subfields.

| Model Name | Backbone | MMLU-Pro | GPQA-D | SuperGPQA | TheoremQA | BBEH |
|---|---|---|---|---|---|---|
| Metric | | Micro | Acc | Macro (discipline) | Acc | Micro |
| MiMo-RL | MiMo-Base | 58.6 | 54.4 | 40.5 | 38.8 | 11.4 |
| QwQ-32B | Qwen2.5-32B-Inst | 52.0 | 54.5 | 43.6 | 48.4 | 22.6 |
| GPT-4o | - | 74.6 | 50.0 | 46.3 | 43.6 | 22.3 |
| o1-mini | - | 80.3 | 60.0 | 45.2 | 53.1 | - |
| DeepSeek-R1 | DeepSeek-V3 | 84.0 | 71.5 | 59.9 | 59.1 | 34.9 |
| *4B Models* | | | | | | |
| Qwen3-4B-Base | - | 51.6 | 26.3 | 25.4 | 34.8 | 8.1 |
| Qwen3-4B-Instruct (non-think) | Qwen3-4B-Base | 61.8 | 41.7 | 32.1 | 42.0 | **14.9** |
| GENERAL-REASONER-4B | Qwen3-4B-Base | **62.8** | **42.9** | **32.5** | **48.3** | 12.2 |
| *7B Models* | | | | | | |
| Qwen2.5-7B-Base | - | 47.7 | 29.3 | 26.7 | 29.1 | 8.0 |
| Qwen2.5-7B-Instruct | Qwen2.5-7B-Base | 57.0 | 33.8 | 30.7 | 36.6 | 12.2 |
| Open-Reasoner-Zero | Qwen2.5-7B-Base | **59.4** | 36.6 | 32.8 | 37.4 | 12.2 |
| Nemotron-CrossThink | Qwen2.5-7B-Base | 57.8 | 38.5 | 29.1 | - | - |
| SimpleRL-Qwen2.5-7B-Zoo | Qwen2.5-7B-Base | 51.5 | 24.2 | 29.9 | 38.0 | 11.9 |
| GENERAL-REASONER-7B | Qwen2.5-7B-Base | 58.9 | **38.8** | **34.2** | **45.3** | **12.5** |
| *14B Models* | | | | | | |
| Qwen2.5-14B-Base | - | 53.3 | 32.8 | 30.7 | 33.0 | 10.8 |
| Qwen2.5-14B-Instruct | Qwen2.5-14B-Base | 62.7 | 41.4 | 35.8 | 41.9 | 15.2 |
| SimpleRL-Qwen2.5-14B-Zoo | Qwen2.5-14B-Base | 64.0 | 39.4 | 35.7 | 40.8 | 13.6 |
| GENERAL-REASONER-Qw2.5-14B | Qwen2.5-14B-Base | **66.6** | **43.4** | **39.5** | **44.3** | **15.2** |
| Qwen3-14B-Base | - | 64.2 | 45.9 | 36.5 | 44.0 | 13.0 |
| Qwen3-14B-Instruct (non-think) | Qwen3-14B-Base | **70.9** | 54.8 | 39.8 | 42.4 | **19.2** |
| GENERAL-REASONER-Qw3-14B | Qwen3-14B-Base | 70.3 | **56.1** | **39.9** | **54.4** | 17.3 |

Table 3: Math task accuracy across datasets. Most evaluations are based on greedy decoding except for AIME24 and AIME25 (averaged over 32 sampling runs following SimpleRL [2]).

| Model Name | AVG | MATH-500 | Olympiad | Minerva | GSM8K | AMC | AIME24 | AIME25 |
|---|---|---|---|---|---|---|---|---|
| *4B Models* | | | | | | | | |
| Qwen3-4B-Base | 40.3 | 68.2 | 34.8 | 42.3 | 72.6 | 47.5 | 10.3 | 6.7 |
| Qwen3-4B-Instruct (non-think) | **54.2** | 80.4 | **49.0** | 57.0 | 92.0 | **62.5** | **22.5** | **16.1** |
| GENERAL-REASONER-4B | 53.4 | **80.6** | 47.7 | **57.7** | **92.2** | 60.0 | 20.0 | 15.4 |
| *7B Models* | | | | | | | | |
| Qwen2.5-7B-Base | 34.7 | 60.2 | 28.6 | 36.0 | 83.1 | 30.0 | 3.8 | 1.4 |
| Qwen2.5-7B-Instruct | 46.3 | 75.0 | 39.4 | 45.2 | 90.9 | 52.5 | 12.5 | 8.5 |
| SimpleRL-Qwen2.5-7B-Zoo | 48.4 | 74.0 | **41.9** | 49.6 | 90.7 | **60.0** | **15.2** | 7.5 |
| GENERAL-REASONER-7B | **48.5** | **76.0** | 37.9 | **54.0** | **92.7** | 55.0 | 13.8 | **10.4** |
| *14B Models* | | | | | | | | |
| Qwen2.5-14B-Base | 37.0 | 65.4 | 33.5 | 24.3 | 91.6 | 37.5 | 3.6 | 2.9 |
| Qwen2.5-14B-Instruct | 49.9 | 77.4 | **44.7** | 52.2 | **94.5** | 57.5 | 12.2 | 11.0 |
| SimpleRL-Qwen2.5-14B-Zoo | 50.7 | 77.2 | 44.6 | 54.0 | 94.2 | 60.0 | 12.9 | 11.8 |
| GENERAL-REASONER-Qw2.5-14B | **53.9** | **78.6** | 42.1 | **58.1** | 94.2 | **70.0** | **17.5** | **16.9** |
| Qwen3-14B-Base | 49.9 | 74.6 | 44.3 | 55.9 | 93.2 | 55.0 | 14.7 | 11.4 |
| Qwen3-14B-Instruct (non-think) | 57.0 | 82.0 | **52.4** | 59.9 | 93.9 | 57.5 | **28.5** | **25.1** |
| GENERAL-REASONER-Qw3-14B | **58.8** | **83.8** | 51.9 | **68.0** | **94.4** | **70.0** | 24.4 | 19.2 |

## 4.3 Baselines

Our main baselines are listed as follows: 1) Qwen2.5 and Qwen3 family [22]: We include these model results to understand the gains of our RL training. 2) SimpleRL [2]: a comprehensive collection of math-based RL models, 3) Open-Reasoner-Zero [33]: a strong RL reasoning model to replicate DeepSeek-R1, which also focused on math training, 4) Nemotron-CrossThink [27]: an all-domain reasoning model. This model is the closet to our general-purpose reasoning model.

### 4.4 Main Results

Table 2 summarizes the performance of our General-Reasoner across general reasoning benchmarks. Overall, General-Reasoner with Zero RL consistently outperforms both base and supervised fine-tuned models across the Qwen2.5 and Qwen3 backbones.

For models initialized with Qwen2.5-7B-Base, General-Reasoner achieves 58.9% on MMLU-Pro, surpassing both the base model at 47.7% and the instructed model at 57.0%. These gains also extend to GPQA and SuperGPQA. Similar improvements appear with the 14B backbone: General-Reasoner-Qw2.5-14B reaches 66.6% on MMLU-Pro, outperforming Qwen2.5-14B-Base at 53.3% and Qwen2.5-14B-Instruct at 62.7%. It also shows strong results on math-related benchmarks, achieving high average scores for both 7B and 14B variants, as shown in Table 3. Compared to other reinforcement learning methods, General-Reasoner consistently outperforms both SimpleRL and Nemotraon-CrossThink across MMLU-Pro, GPQA, SuperGPQA, and BBEH. The trend holds with 14B models, where General-Reasoner achieves the best overall results.

Stronger results are observed when initializing General-Reasoner with Qwen3 backbones. For example, General-Reasoner-4B surpasses Qwen2.5-7B after Zero RL, reaching 62.8% on MMLU-Pro versus 58.9%. This demonstrates the efficiency and transferability of our training method across model scales. The best performing version is General-Reasoner-Qw3-14B, which achieves 56.1% on GPQA and 54.4% on TheoremQA, matching the performance of commercial models like GPT-4o, which scores 50.0% and 43.6% respectively, despite relying solely on Zero RL. Compared to Qwen3-14B-Instruct (non-think), which undergoes post-training via distillation from a much bigger teacher model, our model maintains an advantage on many benchmarks. Notably, even in non-think mode, the Qwen3-Instruct model continues to generate CoT outputs. While some performance gaps remain relative to closed-source or closed-data systems, our results underscore the promise of Zero reinforcement learning when combined with domain-diverse reasoning data and a compact, model-based verifier. Although a performance gap remains on some benchmarks compared to closed-source or closed-data models, our results underscore the promise of Zero reinforcement learning when combined with domain-diverse reasoning data and a compact generative verifier.

Notably, our model does not exhibit overthinking. During training, the average response length grows to around 1,000 tokens, which is significantly shorter than methods like DeepScaleR [4], where outputs can be around 32k tokens. As an example, on the Computer Science split of MMLU-Pro, DeepScaleR-1.5B-Preview requires 18 minutes on 4×H100 GPUs to achieve 35% accuracy. In contrast, our General-Reasoner-4B finishes in just 1.5 minutes with a higher accuracy of 61%.

## 5 Analysis and Ablation Study

### 5.1 Impact of Data Abundance

Table 4: Model performance trained with the diverse domain reasoning data vs. math-only data.

| Backbone | Data | MMLU-Pro | GPQA | SuperGPQA | Math-Related |
|----------|------|----------|------|-----------|--------------|
| Qwen2.5-7B-Base | Full | 58.9 | 34.3 | 34.2 | 48.5 |
| Qwen2.5-7B-Base | Math Only | 56.9 | 32.8 | 29.8 | 49.1 |
| Qwen2.5-14B-Base | Full | 66.6 | 43.4 | 39.5 | 53.9 |
| Qwen2.5-14B-Base | Math Only | 64.8 | 38.9 | 35.6 | 48.6 |

To quantify how domain diversity in the training data affects reasoning performance, we compare two variants of our Zero RL setup: one trained on the full, diverse-domain dataset and one trained exclusively on math-related questions. Table 4 reports results for both Qwen2.5-7B-Base and Qwen2.5-14B-Base backbones under these two conditions.

For the 7B backbone, restricting training to math tasks yields a one point gain on the Math-Related benchmark (49.1% vs. 48.5%), but comes at the expense of general reasoning: MMLU-Pro, GPQA and SuperGPQA each drop by roughly 2 percentage points when compared to the full-data model. In contrast, the model trained on the full dataset achieves stronger performance across all benchmarks, demonstrating more general reasoning capabilities. The 14B backbone exhibits an even clearer benefit from data diversity. With full data, General-Reasoner-Qw2.5-14B outperforms its math-only counterpart on every metric, improving both general benchmarks. These results confirm that

Table 5: Zero RL training using our model-based verifier versus the rule-based verifier on the Qwen3-4B-Base model for 120 step.

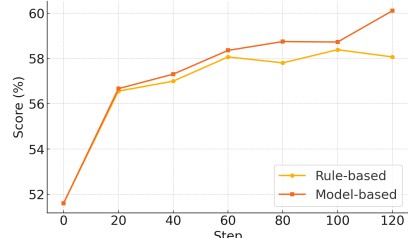

| Dataset | Model-Based | Rule-Based |
|---------|:-----------:|:----------:|
| MMLU-Pro | 60.1 | 58.1 |
| GPQA | 39.4 | 37.9 |
| SuperGPQA | 30.5 | 30.1 |
| Math-Related | 50.4 | 50.0 |

Figure 4: MMLU-Pro evaluation score at different training step using model-based verifier and rule-based verifier.

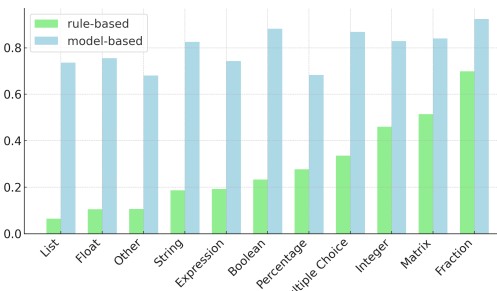

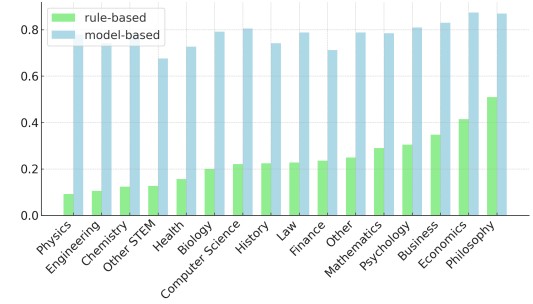

Figure 5: Agreement comparison between rule-based and model-based verifiers against Gemini-2.0-Flash, conditioned on Gemini verifying an answer as correct. Left: Agreement by answer type. Right: Agreement by task subject.

training on diverse reasoning domains enhances general reasoning while maintaining or improving mathematical reasoning.

## 5.2 Impact of General-Verifier on Model Training

We conduct an ablation study to verify the effectiveness of our model-based verifier by comparing models trained using either our model-based verifier or a traditional rule-based verifier[4]. Both setups train the Qwen3-4B-Base model for 120 steps under identical conditions. As shown in Table 5, training with the model-based verifier achieves higher accuracy across all evaluated benchmarks compared to the rule-based verifier.

Figure 4 provides a detailed comparison of performance progression on the MMLU-Pro dataset throughout training. We observe that the rule-based verifier reaches an early plateau at around step 60 (approximately 58%), while the model-based verifier continues improving, achieving about 60% accuracy by step 120. This demonstrates that the model-based verifier effectively guides reinforcement learning, enabling the model to leverage diverse training data better and ultimately achieve stronger overall reasoning performance.

## 5.3 Verifier Agreement with Gemini-2.0-Flash and Human Judgment

To further assess the advantage of the our model-based verifier over the rule-based verifier, we study their agreement with Gemini-2.0-Flash. As discussed earlier, a critical limitation of the rule-based verifier is its high false-negative rate. In a sample of 50k answer-verification pairs that Gemini deemed correct, the rule-based verifier achieved only 22.2% agreement on average, whereas our model-based verifier achieved 78.7%.

We further analyze agreement across different answer types as shown in Figure 5-left. False negatives are most prevalent in unstructured answer types such as Lists or Strings. Even in structured formats like multiple choice, variations like answers being expressed as textual descriptions rather than option letters can cause discrepancies. In Figure 5-right, a domain-level analysis shows that the model-based verifier is particularly beneficial for non-math STEM fields like Physics and Engineering, where

---

[4]The rule-based verifier is implemented based on `https://github.com/huggingface/Math-Verify`

answer formats are diverse. In contrast, fields such as Economics tend to use more structured answers like multiple choices, which narrows the performance gap between the two verifiers.

Table 6: Agreement rates between human judgment and different verifiers over 200 samples, grouped by answer type. Scalar Verifier use the same training data as General-Verifier but predict probability of True/False as scalar value without any chain-of-thought.

| Answer Type | Rule-Based | Gemini-Based | Scalar Verifier | General-Verifier (Ours) |
|---|---|---|---|---|
| Boolean | 0.750 | 1.000 | 1.000 | 1.000 |
| Expression | 0.516 | 0.935 | 0.806 | 0.806 |
| Float | 0.585 | 0.892 | 0.754 | 0.877 |
| Fraction | 0.750 | 1.000 | 1.000 | 1.000 |
| Integer | 0.852 | 0.963 | 0.926 | 0.963 |
| List | 0.556 | 0.778 | 0.778 | 0.889 |
| Matrix | 0.667 | 1.000 | 0.833 | 0.833 |
| Multiple Choice | 0.773 | 0.864 | 0.864 | 0.909 |
| Other | 0.000 | 1.000 | 0.500 | 1.000 |
| Percentage | 0.600 | 1.000 | 0.600 | 0.700 |
| String | 0.312 | 0.938 | 0.812 | 0.812 |

To further substantiate the reliability of our verifiers, we performed a human evaluation of 200 randomly sampled question-answer pairs, comparing the agreement rates of the rule-based, Gemini-based, scalar-based and our General-Verifier against human judgment. As shown in Table 6, our general verifier achieves higher agreement with human annotators than the rule-based and scalar approaches, and is competitive with the Gemini-based verifier across diverse answer types.

## 5.4 Comparison with Supervised Fine-Tuning

Table 7: Comparison between Zero RL training and supervised fine-tuning (SFT) on WebInstruct-Verified.

| Method | MMLU-Pro | SuperGPQA |
|---|---|---|
| *initialize with Qwen3-4B-Base* | | |
| Base | 0.516 | 0.254 |
| w/ WebInstruct SFT | 0.439 | 0.227 |
| w/ WebInstruct-Verified SFT | 0.440 | 0.281 |
| General-Reasoner-Qw3-4B | **0.628** | **0.325** |
| *initialize Qwen3-14B-Base* | | |
| Base | 0.642 | 0.365 |
| w/ WebInstruct SFT | 0.447 | 0.254 |
| w/ WebInstruct-Verified SFT | 0.533 | 0.323 |
| General-Reasoner-Qw3-14B | **0.703** | **0.399** |

We conducted ablation experiments to directly comparing General-Reasoner against models trained via supervised fine-tuning (SFT) on both WebInstruct-Verified and the original WebInstruct datasets. Table 7 reports accuracy on MMLU-PRO and SuperGPQA of the variants, showing that SFT on either WebInstruct-Verified or the original WebInstruct dataset does not reliably improve performance over the base model and may even cause degradation. In contrast, RL-based training yields significant gain, highlighting the effectiveness of our proposed framework.

## 6 Conclusion

In this work, we introduce a training framework for enhancing the general reasoning capabilities of large language models by leveraging a model-based verifier to scale verifiable training data across diverse domains. By applying GRPO training directly to base LLMs using our curated, high-quality, verifiable dataset and a generative model-based verifier, we demonstrate competitive reasoning performance against models that require an additional supervised fine-tuning stage. Our approach achieves strong generalization across various challenging domains while preserving superior effectiveness in mathematical reasoning tasks.

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

# A  Appendix

## A.1  Limitation

This work focuses on the Zero RL setting, where models are trained directly from base language models without intermediate supervised fine-tuning. However, the proposed integration of model-based answer verification and diverse-domain reasoning data is not tied to Zero RL. Exploring its impact when combined with intermediate stages, such as supervised fine-tuning or distillation is promising to have further performance gain.

Additionally, our study targets general language reasoning tasks across diverse domains. We do not specifically focus on code reasoning or olympiad-level math competitions. Integrating these specialized reasoning domains into our framework is a valuable avenue for future extension.

In this work, the General-Verifier focuses on verifying short answers. Future work could explore extending it to long-form answers and further investigate the risks associated with reward hacking.

## A.2  Broader Impact

This work improves the general reasoning capabilities of large language models, making them more applicable to real-world scenarios where complex, multi-domain reasoning is essential. By enabling LLMs to go beyond math and code, our approach allows AI systems to better support humans in tasks such as decision-making, analysis, and problem-solving across diverse fields, ultimately increasing their utility in production settings and high-impact applications.

## A.3  Detailed Results

We provide detailed evaluation results on MMLU-pro, SuperGPQA and BBEH are listed in below. Following previous work, the average reported in MMLU-Pro is micro-average, and for SuperGPQA and BBEH, we report macro-average across subtask/domain.

Table 8: Per-domain accuracy comparison of different models on MMLU-Pro.

| Model Name | Avg | CS | Math | Chem | Eng | Law | Bio | Health | Phys | Bus | Phil | Econ | Other | Psy | Hist |
|---|---|---|---|---|---|---|---|---|---|---|---|---|---|---|---|
| Qwen3-4B-Base | 51.6 | 49.3 | 67.8 | 59.6 | 44.8 | 26.0 | 67.2 | 49.1 | 58.1 | 59.2 | 33.1 | 58.5 | 40.3 | 53.8 | 34.4 |
| GENERAL-REASONER-4B | 62.8 | 61.7 | 81.6 | 69.9 | 50.9 | 29.2 | 79.8 | 61.0 | 70.0 | 71.4 | 48.9 | 71.1 | 54.3 | 66.3 | 47.8 |
| Qwen2.5-7B-Base | 47.7 | 51.7 | 58.9 | 45.1 | 34.5 | 25.5 | 64.6 | 46.6 | 50.0 | 55.4 | 34.9 | 57.1 | 47.6 | 54.8 | 39.1 |
| Qwen2.5-7B-Instruct | 57.0 | 57.1 | 71.4 | 57.3 | 43.1 | 31.3 | 71.4 | 55.4 | 60.7 | 64.8 | 44.9 | 67.1 | 54.8 | 63.0 | 48.3 |
| SimpleRL-Qwen2.5-7B-Zoo | 51.5 | 54.9 | 55.4 | 48.7 | 40.9 | 30.8 | 68.3 | 53.1 | 53.4 | 58.0 | 41.3 | 62.6 | 52.2 | 60.7 | 42.5 |
| GENERAL-REASONER-7B | 58.9 | 61.5 | 75.5 | 62.1 | 48.1 | 32.1 | 71.7 | 57.8 | 62.4 | 66.2 | 44.3 | 67.2 | 54.3 | 63.5 | 47.0 |
| Qwen2.5-14B-Base | 53.3 | 54.6 | 63.0 | 52.8 | 36.0 | 31.9 | 71.3 | 56.5 | 52.9 | 61.1 | 46.1 | 64.8 | 50.1 | 61.0 | 44.4 |
| Qwen3-14B-Base | 64.2 | 70.2 | 77.4 | 67.3 | 48.7 | 33.2 | 80.8 | 65.5 | 68.7 | 70.7 | 53.5 | 74.4 | 60.7 | 69.7 | 54.6 |
| Qwen2.5-14B-Instruct | 62.7 | 66.6 | 75.3 | 63.0 | 39.7 | 37.4 | 79.6 | 65.2 | 63.9 | 69.3 | 53.5 | 72.0 | 63.4 | 72.1 | 59.1 |
| SimpleRL-Qwen2.5-14B-Zoo | 64.0 | 66.1 | 75.8 | 66.9 | 49.8 | 37.2 | 79.5 | 64.1 | 67.7 | 69.5 | 55.5 | 73.8 | 61.4 | 70.9 | 53.8 |
| GENERAL-REASONER-Qw2.5-14B | 66.6 | 69.8 | 78.8 | 67.5 | 54.8 | 39.7 | 81.7 | 65.3 | 71.4 | 71.9 | 56.7 | 74.4 | 64.0 | 73.2 | 59.1 |
| GENERAL-REASONER-Qw3-14B | 70.3 | 73.9 | 86.5 | 76.4 | 55.5 | 39.9 | 83.3 | 70.5 | 76.1 | 76.6 | 58.3 | 78.7 | 66.2 | 73.3 | 58.0 |

Table 9: Per-domain accuracy comparison of different models on SuperGPQA.

| Model Name | Avg. | Eng. | Med. | Sci. | Phil. | Mil. Sci. | Econ. | Mgmt. | Socio. | Lit./Arts | Hist. | Agron. | Law | Edu. |
|---|---|---|---|---|---|---|---|---|---|---|---|---|---|---|
| Qwen3-4B-Base | 25.4 | 27.3 | 26.1 | 26.7 | 26.5 | 23.9 | 30.7 | 29.3 | 23.1 | 19.9 | 15.3 | 23.1 | 29.3 | 28.7 |
| GENERAL-REASONER-4B | 32.5 | 34.6 | 32.9 | 34.5 | 38.0 | 35.1 | 37.8 | 33.9 | 36.4 | 24.4 | 19.9 | 31.1 | 29.4 | 34.9 |
| Qwen2.5-7B-Base | 26.7 | 25.1 | 26.7 | 23.8 | 29.7 | 28.8 | 29.2 | 31.7 | 28.0 | 21.5 | 18.2 | 25.6 | 27.7 | 31.6 |
| Qwen2.5-7B-Instruct | 30.7 | 29.2 | 31.2 | 27.9 | 32.3 | 36.1 | 32.9 | 33.7 | 36.4 | 24.8 | 20.5 | 27.4 | 31.3 | 35.1 |
| SimpleRL-Qwen2.5-7B-Zoo | 29.9 | 28.0 | 31.3 | 26.0 | 34.9 | 32.2 | 32.7 | 31.5 | 34.3 | 25.0 | 23.1 | 27.2 | 29.4 | 33.5 |
| GENERAL-REASONER-7B | 34.2 | 32.3 | 34.5 | 31.1 | 36.3 | 42.4 | 38.3 | 36.5 | 41.3 | 25.1 | 23.3 | 29.5 | 34.2 | 39.9 |
| Qwen2.5-14B-Base | 30.7 | 29.2 | 31.2 | 27.9 | 32.3 | 36.1 | 32.9 | 33.7 | 36.4 | 24.8 | 20.5 | 27.4 | 31.3 | 35.1 |
| Qwen3-14B-Base | 36.5 | 36.1 | 38.8 | 34.4 | 37.8 | 44.4 | 41.8 | 40.3 | 38.5 | 28.8 | 26.7 | 33.6 | 36.4 | 37.2 |
| Qwen2.5-14B-Instruct | 35.8 | 35.6 | 37.1 | 34.1 | 38.6 | 36.1 | 41.8 | 39.5 | 39.2 | 30.7 | 26.6 | 32.2 | 36.1 | 37.4 |
| SimpleRL-Qwen2.5-14B-Zoo | 35.7 | 34.2 | 36.7 | 33.0 | 36.0 | 40.0 | 39.9 | 41.3 | 38.5 | 30.8 | 26.7 | 31.1 | 38.1 | 38.2 |
| GENERAL-REASONER-Qw2.5-14B | 39.5 | 36.5 | 41.6 | 35.6 | 41.8 | 41.9 | 44.7 | 42.9 | 42.7 | 32.8 | 29.7 | 37.9 | 39.6 | 45.3 |
| GENERAL-REASONER-Qw3-14B | 39.9 | 42.6 | 44.9 | 42.3 | 41.5 | 45.9 | 47.0 | 42.7 | 41.9 | 29.2 | 28.2 | 34.5 | 34.5 | 43.0 |

Table 10: Per sub-task accuracy comparison of different models on BBEH.

| Model | Avg | BGQ | Bool | Bug | Caus | Disam | Dyck | Geom | Hyp | Ling | Movie | MultiA | NYCC | ObjC | ObjP | Sarc | Shuf | Spat | Sport | Temp | TimeA | Web | Sort | Zebra |
|---|---|---|---|---|---|---|---|---|---|---|---|---|---|---|---|---|---|---|---|---|---|---|---|---|
| Qwen3-4B-Base | 8.1 | 36.0 | 12.5 | 1.5 | 41.5 | 18.3 | 0.0 | 3.5 | 0.5 | 1.5 | 28.5 | 0.0 | 10.0 | 0.5 | 1.5 | 6.0 | 3.0 | 0.5 | 10.5 | 2.5 | 11.0 | 8.0 | 0.5 | 9.5 |
| General-Reasoner-4B | 12.2 | 33.5 | 14.0 | 1.0 | 47.0 | 40.8 | 4.5 | 23.5 | 0.5 | 4.0 | 39.0 | 5.0 | 12.5 | 0.0 | 0.0 | 16.0 | 1.5 | 7.0 | 16.5 | 3.0 | 19.0 | 12.0 | 6.5 | 0.5 |
| Qwen2.5-7B-Base | 8.0 | 26.5 | 10.0 | 0.0 | 39.0 | 18.3 | 3.0 | 9.5 | 1.0 | 1.5 | 30.0 | 0.0 | 6.0 | 0.0 | 0.5 | 11.5 | 2.0 | 5.0 | 3.5 | 4.5 | 11.0 | 5.5 | 0.5 | 16.5 |
| Qwen2.5-7B-Inst | 12.2 | 36.0 | 5.5 | 0.5 | 48.5 | 37.5 | 0.5 | 24.5 | 1.5 | 4.0 | 37.0 | 0.0 | 9.0 | 0.0 | 3.5 | 17.0 | 9.0 | 8.0 | 15.5 | 0.5 | 21.0 | 9.0 | 2.5 | 16.0 |
| SimpleRL-7B-Zoo | 11.9 | 29.5 | 9.5 | 0.5 | 45.0 | 44.2 | 0.5 | 27.0 | 0.0 | 3.0 | 28.5 | 1.0 | 9.5 | 0.0 | 2.0 | 16.5 | 8.5 | 8.0 | 12.0 | 3.0 | 16.0 | 14.0 | 5.5 | 14.5 |
| General-Reasoner-7B | 12.5 | 28.5 | 15.5 | 1.0 | 42.5 | 45.8 | 2.5 | 19.5 | 0.0 | 3.5 | 34.5 | 1.0 | 12.0 | 0.0 | 1.5 | 18.0 | 8.5 | 9.5 | 15.0 | 1.5 | 18.5 | 8.0 | 4.0 | 22.0 |
| Qwen2.5-14B-Base | 10.8 | 34.5 | 13.0 | 1.5 | 48.5 | 41.7 | 0.5 | 9.0 | 2.5 | 3.0 | 31.0 | 0.5 | 11.5 | 0.0 | 0.0 | 17.0 | 4.5 | 0.5 | 10.0 | 1.0 | 16.0 | 8.0 | 6.0 | 15.0 |
| Qwen2.5-14B-Base | 13.0 | 36.0 | 10.5 | 1.0 | 40.5 | 45.0 | 5.0 | 19.5 | 2.5 | 5.0 | 41.0 | 3.5 | 12.0 | 0.0 | 1.0 | 25.0 | 7.0 | 5.5 | 14.0 | 1.5 | 19.5 | 9.0 | 6.5 | 13.5 |
| Qwen2.5-14B-Inst | 15.2 | 43.5 | 22.5 | 1.5 | 50.5 | 50.0 | 2.0 | 39.0 | 9.0 | 7.0 | 43.0 | 3.0 | 13.5 | 0.0 | 2.0 | 29.0 | 6.0 | 10.0 | 19.5 | 2.0 | 28.5 | 13.0 | 4.5 | 19.0 |
| SimpleRL-14B-Zoo | 13.6 | 29.0 | 7.0 | 0.0 | 50.0 | 48.3 | 4.0 | 33.0 | 1.0 | 3.5 | 42.5 | 0.5 | 13.5 | 0.0 | 2.5 | 25.0 | 1.0 | 8.5 | 19.5 | 2.0 | 23.5 | 13.0 | 8.0 | 6.0 |
| General-Reasoner-Qw2.5-14B | 15.2 | 35.0 | 24.0 | 0.0 | 50.5 | 45.8 | 2.5 | 22.5 | 0.5 | 6.0 | 53.5 | 4.5 | 14.0 | 1.5 | 2.0 | 22.0 | 7.5 | 13.0 | 18.5 | 2.0 | 10.5 | 18.0 | 9.0 | 16.5 |
| General-Reasoner-Qw3-14B | 17.3 | 33.0 | 21.5 | 2.0 | 49.5 | 53.3 | 4.0 | 24.5 | 2.5 | 9.0 | 49.0 | 18.5 | 10.5 | 0.5 | 1.5 | 25.5 | 10.0 | 13.5 | 22.0 | 2.0 | 43.5 | 17.0 | 8.5 | 6.5 |

## A.4 Verifier Input-Output Example

**Verifier Input:**
User: ### Question: Solve the Bernoulli equation

$$x\frac{dy}{dx} + y = x^2y^2$$

### Ground Truth Answer:

$$\frac{1}{y} = -x^2 + Cx$$

### Student Answer:

$$y = \frac{1}{Cx - x^2}$$

For the above question, please verify if the student's answer is equivalent to the ground truth answer. You don't need to redo the question yourself, just check if the student's answer equivalent to the ground truth answer. If the student's answer is correct, output "Final Decision: Yes". If the student's answer is incorrect, output "Final Decision: No".

**Verifier Output:**
Assistant:
1. Ground Truth Answer:

$$\frac{1}{y} = -x^2 + Cx$$

2. Student Answer:

$$y = \frac{1}{Cx - x^2}$$

We can rewrite the ground truth answer as:

$$\frac{1}{y} = Cx - x^2$$

$$y = \frac{1}{Cx - x^2}$$

Comparing this with the student's answer, we see that they are the same.
Final Decision: Yes

## A.5 Detailed Hyper-Parameters

We provide the detailed hyperparameters for training our General-Reasoner variants in Table 11. The difference in batch size configurations between the Qwen2.5 and Qwen3 series is due to the limitations in earlier versions of vLLM (e.g., v0.6.3), which did not support engine sleep and wake mechanisms for properly loading and unloading the verifier model. As a result, we had to dedicate 2 GPUs to the verifier and 6 GPUs to policy model for each node. With the newer vLLM version (v0.8.5), we are able to share GPU resources more efficiently by leveraging proper parameter loading and unloading.

Table 11: Hyperparameter settings for General-Reasoner variants.

| parameter | GENERAL-REASONER-7B | GENERAL-REASONER-Qw2.5-14B | GENERAL-REASONER-4B | GENERAL-REASONER-Qw3-14B |
|---|---|---|---|---|
| n_nodes | 4 | 4 | 4 | 4 |
| n_gpu | 8×H100 | 8×H100 | 8×H100 | 8×H100 |
| backbone_hgf_id | Qwen/Qwen2.5-7B | Qwen/Qwen2.5-14B | Qwen/Qwen3-4B-Base | Qwen/Qwen3-14B-Base |
| vllm_version | 0.6.3 | 0.6.3 | 0.8.5 | 0.8.5 |
| train_batch_size | 768 | 768 | 1024 | 1024 |
| max_prompt_length | 1024 | 1024 | 1024 | 1024 |
| max_response_length | 4096 | 4096 | 4096 | 4096 |
| learning_rate | 5e-7 | 5e-7 | 5e-7 | 5e-7 |
| ppo_mini_batch_size | 192 | 192 | 256 | 256 |
| ppo_micro_batch_size | 4 | 4 | 4 | 4 |
| clip_ratio_low | 0.3 | 0.3 | 0.2 | 0.2 |
| clip_ratio_high | 0.3 | 0.3 | 0.3 | 0.3 |
| kl_loss_coef | 0.0001 | 0.0001 | 0.0001 | 0.0001 |
| kl_loss_type | low_var_kl | low_var_kl | low_var_kl | low_var_kl |
| temperature | 1.0 | 1.0 | 0.7 | 0.7 |
| rollout_n | 8 | 8 | 8 | 8 |
| kl_coef | 0.001 | 0.001 | 0.001 | 0.001 |

