# OpenReview forum: "General-Reasoner: Advancing LLM Reasoning Across All Domains"
_NeurIPS.cc/2025/Conference — NeurIPS 2025 poster_

### Official Review · Reviewer_nfc4 · 2025-07-01

**Clarity:** 2
**Significance:** 2
**Originality:** 1
**Rating:** 3
**Confidence:** 4

**Summary:**

This paper introduces General-Reasoner, a novel training paradigm designed to enhance the reasoning capabilities of large language models (LLMs) across diverse domains. It addresses the limitations of existing methods, which primarily focus on mathematical and coding tasks. The authors construct WebInstruct-verified, a large-scale, high-quality dataset spanning multiple disciplines, and develop General-Verifier, a generative model-based verifier that replaces rigid rule-based verification. This approach enables robust reward signals for reinforcement learning (RL).
Comprehensive evaluations demonstrate that General-Reasoner outperforms baseline methods on general reasoning benchmarks such as MMLU-Pro and SuperGPQA, while maintaining strong performance in mathematical tasks. The work contributes a scalable framework for improving LLM reasoning across a broad range of domains.

**Questions:**

1. Clarify the innovation of the General-Verifier: Specifically, how does it differ from existing verifiers in terms of architecture, training, or performance?
2. Extend training steps in Figure 4: The sharp performance rise at step 120 may be coincidental. Could you train for more steps (e.g., 200+) or until convergence to validate robustness?
3. Separate Qwen3-14B and Qwen2.5 results: Grouping them together may obscure performance differences. Please disaggregate the results for fair comparison.
4. Expand baseline comparisons: Include more results for Qwen3-14B baselines to better contextualize General-Reasoner’s improvements.

typos: There is a typo in the OpenReview abstract where '\model' is not rendered correctly.

**Ethical Concerns:**

["NO or VERY MINOR ethics concerns only"]

**Final Justification:**

Based on the author's rebuttal, points 3 and 4 have been resolved, so I have decided to raise my overall rating.

**Limitations:**

yes

**Quality:**

2

**Strengths And Weaknesses:**

## Strengths

1. The paper introduces a novel model-based verifier that significantly improves answer verification across diverse domains, addressing a key limitation of rule-based methods.
2. The construction of a large-scale, high-quality dataset (WebInstruct-verified) enhances the scope of reasoning tasks beyond mathematics, contributing valuable resources for future research.
3. Comprehensive evaluations across multiple benchmarks demonstrate the model's robust performance and generalization capabilities, particularly in non-mathematical domains.

## Weaknesses

1. The use of a model-based verifier (or reward model in RL) to replace rule-based verification is not novel, as evidenced by prior works like [1][2][3], and has been applied even in math-focused tasks.
2. GRPO already employs model-based reward systems (as shown in [4], Table 10), making the proposed modification less innovative.
3. Table 4 shows only modest improvements on MMLU and GPQA when using diverse data versus math-only data, and the comparisons with Qwen3-14B lack strong baselines, raising doubts about the method's impact.
4. The answer-pair verification approach is limited to QA-style tasks with short answers, failing to generalize to broader tasks like summarization, writing, or translation.

[1] VerifyBench: Benchmarking Reference-based Reward Systems for Large Language Models

[2] xVerify: Efficient Answer Verifier for Reasoning Model Evaluations

[3] Compassjudger-1: All-in-one judge model helps model evaluation and evolution

[4] DeepSeekMath: Pushing the Limits of Mathematical Reasoning in Open Language Models

---

> ### Author Rebuttal · Authors · 2025-07-31
>
> We sincerely thank the reviewer for highlighting the contribution of our datasets and the effectiveness of the model-based verifier in RLVR training. We also appreciate the insightful comments and valuable feedback. Below is our point-by-point response addressing each comment raised:
>
> 1. **The use of a model-based verifier (or reward model in RL) to replace rule-based verification is not novel, as evidenced by prior works like [1][2][3], and has been applied even in math-focused tasks.**
>
>     The works mentioned, such as VerifyBench [1] and xVerify [2], are very close concurrent works to ours, published within 3 months of the submission deadline. Their preprints were actually released after the first version of our preprint.
>
>     Beyond timing considerations, our work's main contribution is scaling RLVR data across diverse domains, with the generative model-based verifier serving as a solution that effectively supports this data scaling. This differentiates our approach from previous works (e.g., [3]) that only focus on improving answer verification in the math domain. Our work provides a scalable framework that advancing the field beyond narrow, domain-specific reasoning enhancements.
>
> 2. **GRPO already employs model-based reward systems**
>
>     While model-based reward systems aren't entirely novel, our work specifically focuses on developing a lightweight, generative model-based verifier that enables scaling across diverse domains.
>
> 3. **The sharp performance rise at step 120 may be coincidental.**
>
>     We thank the reviewer for raising this question. To further investigate, we retrained the model-verifier-based General-Reasoner-Qwen3-4B corresponding to Figure 4 for 500 steps, with the results shown below:
>
>     | Steps | MMLU-PRO |
>     | --- | --- |
>     | 40 | 0.5767 |
>     | 80 | 0.5865 |
>     | 120 | 0.5958 |
>     | 160 | 0.6061 |
>     | 200 | 0.6140 |
>     | 240 | 0.6082 |
>     | 280 | 0.6173 |
>     | 320 | 0.6142 |
>     | 360 | 0.6239 |
>     | 400 | 0.6234 |
>     | 440 | 0.6319 |
>     | 480 | 0.6312 |
>
>     This demonstrates that the performance improvement with the model-based verifier is consistent and not coincidental.
>
> 4. **Expand baseline comparisons: Include more results for Qwen3-14B baselines**
>
>     Our main baselines in this work are the ZeroRL approaches such as SimpleRL. However, since Qwen3-14B is a newly released model, no baselines from previous works are available for it. The comparisons with ZeroRL baselines focus on the Qwen2.5 models, while our Qwen3 experiments demonstrate that our data and training framework also bring improvements to these recent stronger models.
>
>     To further enrich the baseline comparison, we trained two SFT baselines: 1) we directly used WebInstruct-Verified data to conduct SFT with questions and extracted short answers, and 2) we used the original WebInstruct (without processing it as verifiable answer data with open answer form) to conduct SFT. The results are shown below:
>
>     | Model | Post Training | MMLU-PRO | SuperGPQA |
>     | --- | --- | --- | --- |
>     | Qwen3-4B-Base | - | 0.516 | 0.254 |
>     | Qwen3-4b-webinstruct-verified-sft | SFT | 0.4397 | 0.281 |
>     | Qwen3-4b-webinstruct-sft | SFT | 0.4386 | 0.227 |
>     | General-Reasoner-Qw3-4B | RL | 0.628 | 0.325 |
>     |  |  |  |  |
>     | Qwen3-14B-Base | - | 0.642 | 0.365 |
>     | Qwen3-14b-webinstruct-verified-sft | SFT | 0.5332 | 0.323 |
>     | Qwen3-14b-webinstruct-sft | SFT | 0.4465 | 0.2542 |
>     | General-Reasoner-Qw3-14B | RL | 0.703 | 0.399 |
>
>     The results demonstrate that SFT training on strong base models actually degrades performance. In contrast, our RL-based training framework consistently delivers significant improvements over the base model.
>
> 5. **Separate Qwen3-14B and Qwen2.5 results**
>
>     We appreciate your suggestion and will separate the Qwen3 and Qwen2.5 results for the 14B models into different groups to enhance clarity in the presentation.
>
> 6. **The answer-pair verification approach is limited to QA-style tasks with short answers, failing to generalize to broader tasks like summarization, writing, or translation.**
>
>     This work focuses on improving LLM reasoning capabilities specifically within the scope of reinforcement learning with verifiable reward (RLVR) for reasoning-intensive tasks. Tasks like summarization, writing, and translation are not reasoning-intensive and fall outside the scope of our research. We will expand this discussion in the limitations section.
>
> 7. **typo in OpenReview abstract**
>
>     We thank the reviewer for their careful reading. We will fix the typo.
>
>
> We hope our response has addressed your main questions and concerns. We welcome further discussion during the reviewer-author discussion period.

---

> > ### Comment · Area_Chair_MnT5 · 2025-08-08
> >
> > Dear Reviewer,
> >
> > Today is the final day of the author–reviewer discussion period.
> > Reviewers are expected to engage with the authors to acknowledge the rebuttal and actively contribute to the discussion to clarify any open points. Thank you for your timely participation in this process.

---

### Official Review · Reviewer_JeZ6 · 2025-07-02

**Clarity:** 3
**Significance:** 2
**Originality:** 2
**Rating:** 4
**Confidence:** 4

**Summary:**

This paper tries to extend RL training of LLMs beyond verifiable domains like math and coding where rule-based rewards can be applied.

To do this, they first constructed a general-domain training set by filtering WebInstruct-verified, and then trained a small generative reward model as the answer verifier for the general-domain QA task so that computing the reward can be inefficient. The small generative reward model is based on Qwen2.5-Math-1.5B and distilled from Gemini-2.0. They then trained the model with standard GRPO (without any warmstart with SFT).

The trained models (named General-Reasoner) gets decent improvement as compared to the Base models on MMLU-Pro, GPQA, SuperGPQA, and BBEH in general. And in fact, they get improvement on math benchmarks too probably due to the learned reasoning behaviour.

**Questions:**

- What's the SFT data for "Qwen3-4B-Instruct" in the second block of Table 2? Is that just one of the official relased checkpoints from Qwen?

**Ethical Concerns:**

["NO or VERY MINOR ethics concerns only"]

**Final Justification:**

The additional comparison of SFT vs RL on WebInstruct (SFT version vs RL version of the dataset) is interesting.
Also, I did not realize the fact that some of the related works are released AFTER this paper.
I raised my score to 4 accordingly.

**Quality:**

3

**Strengths And Weaknesses:**

Weakness:

- I think at this point we all agree that Qwen models are outliers in the sense that their mid-training probably made them extremely suitable for RL. So I do think it's a fair ask to request additional experiments on non-Qwen checkpoints to verifiy the generalizability of the proposed method.

- You might want to justify the novelty of the work a bit more: For example, in "Demystifying Long Chain-of-Thought Reasoning in LLMs", there are also RL experiments on general-domain QA data (Section 5) and they also used WebInstruct data. What's the fundamental difference apart from your own small verifier model?

Strength:

- Thorough experiments and a trendy topic
- Good empirical results

---------------------------------------------------------------
Update on 7/4: I was prompted by the AC to add more reasons for rejection, so I'm adding more thoughts here. Quite frankly, I think the paper is offering some interesting empirical results for the RLVR community, and the execution is generally ok; it just doesn't quite hit the scientific standard I was hoping for in a NeurIPS paper. I will share some more thoughts/questions that will hopefully be helpful:

The paper has two main contributions: (1) curating a general-domain training set for doing RLVR beyond math tasks; and (2) training a small model-based answer verifier as the reward model in RL, rather than using rule-based rewards.

For (1):

- In your filtering step, you mentioned you only kept examples where you can find human-written answers in the original websites. How exactly was this step implemented? Did you use an LLM filter to check the original websites?

- Quite frankly, a significant portion of the dataset is still math problems (34%).

- How do you justify the quality of the final dataset? And how do you justify its necessity? Do we really need RL data on general-domain questions? Because if you look at the paper "Does Math Reasoning Improve General LLM Capabilities? Understanding Transferability of LLM Reasoning", in their Figure 1, even if you RL on math data, you still get improvement on non-math tasks. If you wanna claim this new training dataset is a contribution, shouldn't you do some data ablation to justify its importance? Does training on your filtered WebInstruct data offer additional gains beyond simply training on existing math data?

For (2):

- I like Table 5 in the sense that it's clearly telling me what's the advange of model-based verifier vs rule-based verifier. It'd be nice if there could be some results of using an "oracle" reward too. For eg, by using Gemini as the verifier there. In a way, I really like the various ablations being done in the paper "Pitfalls of Rule- and Model-based Verifiers – A Case Study on Mathematical Reasoning". They showed clearly the gains and potential reward hacking of using model-based verifiers. Do you think your work is offering additional new insights beyond their results? (Apart from experimenting it on different datasets?)

In general, I would happily raise my score if you could tell me concretely the list of scientific insights that your paper is offering beyond what's already known by previous RLVR papers. For to give you some few-shot examples, here are some examples extracted from the papers I mentioned above:

- RL training on math can get transferable improvement on other non-math benchmarks or even non-reasoning benchmarks (from "Does Math Reasoning Improve General LLM Capabilities? Understanding Transferability of LLM Reasoning")
- Using model-based verifier in RL training can be better than rule-based verifiers, but in some cases, there is reward-hacking (from "Pitfalls of Rule- and Model-based Verifiers – A Case Study on Mathematical Reasoning".
- You can do RL on general-domain data (WebInstruct) with either model-based verifier or rule-based verifier, ideally on a filtered set of prompts with short answers, and it can get improvement on general benchmarks like MMLU-Pro (literally Takeaway 5.2 from "Demystifying Long Chain-of-Thought Reasoning in LLMs").

In general, I was hoping for a scientific paper beyond "we gathered a new training dataset by filtering an existing dataset, and finetuned a new model with existing training algorithms, and it works slightly better on some benchmarks". Or at least this shouldn't be the narrative of the paper IMHO. Happy to discuss more during rebuttal!

---

> ### Author Rebuttal · Authors · 2025-07-31
>
> We sincerely thank the reviewer for highlighting our empirical results and our contribution in scaling data for RLVR training across diverse domains. We also appreciate the valuable feedback and comments. Below is our point-by-point response addressing each comment raised:
>
> 1. **“What's the SFT data for "Qwen3-4B-Instruct" in the second block of Table 2? Is that just one of the official relasedcheckpoints from Qwen?”**
>
>     Yes, the Qwen3-4B-Instruct baseline is one of the official released checkpoints from Qwen. We will add further clarification in the experiment setup.
>
> 2. **“How exactly was [kept examples where you can find human-written answers] implemented?”**
>
>     Thanks for raising this question, here what we trying to say is not guarantee “human” answer in contrast to “synthetic” answers. As the source data is from webcrawling rather than LLM generated, it is naturally “human-written” contents, however, we use Gemini-Pro to ensure that the completeness of human-written answers for the extracted questions. We will update the description to improve the clarity.
>
> 3. **“a significant portion of the dataset is still math problems (34%). “**
>
>     While our dataset encompasses diverse reasoning domains, we maintained a significant portion (34%) focused on math problems. This decision is based on the fact that many STEM-domain tasks depend on mathematical reasoning abilities. Nevertheless, we highlight that 66% of our dataset covers non-mathematical domains, which represents a substantial departure from existing works that focused only on mathematical reasoning.
>
> 4. **“How do you justify the quality of the final dataset? shouldn't you do some data ablation to justify its importance? “**
>
>     In Table 4, we compare performance when using our full training dataset versus a math-only subset while keeping the verifier fixed. The results demonstrate the effectiveness of scaling training data across multiple reasoning domains. Our method also outperforms previous approaches like SimpleRL, which primarily focuses on math-only data, further validating the importance of incorporating diverse reasoning tasks.
>
> 5. **“insights beyond existing / concurrent works”**
>
>     1. **Demystifying Long Chain-of-Thought Reasoning in LLMs**
>
>         *Preprint Date: Feb 5th, 2025 (concurrent work within 3 month)*
>
>         We would like to clarify that the WebInstruct dataset used in the referenced work was designed for supervised fine-tuning (SFT). In contrast, our WebInstruct-Verified dataset specifically targets high-quality RLVR training data. To demonstrate the effectiveness of our approach, we conducted additional experiments comparing SFT training with WebInstruct against RL training using WebInstruct-Verified on Qwen3-4B and Qwen3-14B.
>
>         | Model | Post Training | MMLU-PRO | SuperGPQA |
>         | --- | --- | --- | --- |
>         | Qwen3-4B-Base | - | 0.516 | 0.254 |
>         | qwen3-4b-webinstruct-sft | SFT | 0.4386 | 0.227 |
>         | General-Reasoner-Qw3-4B | RL | 0.628 | 0.325 |
>         |  |  |  |  |
>         | Qwen3-14B-Base | - | 0.642 | 0.365 |
>         | Qwen3-14b-webinstruct-sft | SFT | 0.4465 | 0.2542 |
>         | General-Reasoner-Qw3-14B | RL | 0.703 | 0.399 |
>
>         The results demonstrate that using our curated data with the proposed training framework is more effective than applying SFT over WebInstruct. Based on these findings, we conclude the following takeaway:
>
>         **Insight**: *The curation of extract verifiable question-answer pair to create diverse domain RLVR training data is more effective than directly apply on crawled instruction following data such as WebInstruct.*
>
>     2. **Pitfalls of Rule- and Model-based Verifiers – A Case Study on Mathematical Reasoning**
>
>         *Preprint Date: May 28th, 2025 (follow up work of ours)*
>
>         This referenced work is actually a follow-up study to ours, and notably, our verifier model was extensively employed in their experimental comparisons. Their experiments validate that our high-quality model-based verifier performs well compared to other verifiers. We highly appreciate their additional analyses and insights.
>
>         Our primary contribution is scaling Zero-RL training across general domains and evaluating the broader improvements in reasoning capabilities of LLMs. While an extensive characterization of model-based verifiers is beyond the scope of our current work, especially since their study specifically addresses mathematical reasoning, we aim to clearly deliver this key takeaway:
>
>         ***Insight**: Model-based verifier have important role in help scaling the RLVR training data across diverse domains.*
>
>     3. **Does Math Reasoning Improve General LLM Capabilities? Understanding Transferability of LLM Reasoning**
>
>         *Preprint Date: July 1th, 2025 (follow up work of ours)*
>
>         Their work includes General-Reasoner in its major evaluation, which demonstrates strong transferability. We find their analyses of transferability particularly impressive.
>
>         Returning to our research, our findings align with evidence that mathematical reasoning enhances general LLM capabilities. However, we demonstrate significant additional benefits from scaling RL training to include diverse reasoning tasks. Comparing SimpleRL-14B with Qwen2.5-14B-Base on MMLU-PRO shows that math-focused RL improves general reasoning performance. Our General-Reasoner-Qwen2.5-14B further surpasses these results, highlighting the substantial gains from broader-domain reasoning training. Additionally, Table 4 demonstrates the improvements achieved by incorporating diverse data compared to using math-only data.
>
>         ***Insight**: While math reasoning enhances general LLM capabilities, scaling training to include diverse reasoning domains significantly increases effectiveness.*
>
>
>     We sincerely thank the reviewer for bringing up these comparisons, which enrich the discussion of our paper. We hope that the above analysis of the suggested comparisons clearly demonstrates the specific contributions and insights our work provides.

---

### Official Review · Reviewer_MFcj · 2025-07-03

**Clarity:** 3
**Significance:** 2
**Originality:** 2
**Rating:** 4
**Confidence:** 3

**Summary:**

General-Reasoner introduces a zero-RL fine-tuning framework that leverages a newly curated, 230 K-question, multi-domain reasoning dataset (“WebInstruct-verified”) and a 1.5 B-param generative verifier to provide reward signals across diverse answer formats; by directly applying policy optimization to base LLMs (Qwen2.5-7B/14B and Qwen3-4B/14B), it achieves consistent ~10 perc gains over baselines on 12 STEM and humanities benchmarks.

**Questions:**

1. Did the authors compare the verifier against Gemini generated answers?
2. What if the verifier is trained only with open LLMs? All curation/verifier training described uses Gemini, no ablation with only open-source models.
3. Is generative verifier better than scalar RLVR?
4. Over-reliance on a single verifier model backbone: are there any ablations on alternative verifiers?

**Ethical Concerns:**

["NO or VERY MINOR ethics concerns only"]

**Final Justification:**

Still, my perspective is that the paper’s main contribution lies in the dataset curation. While I recognize its value, I believe the work aligns more naturally with the scope of the dataset track. As such, I find it less compelling as a main track paper. My borderline accept rating reflects this—acknowledging the merit of the dataset, but with reservations about its fit for the main track.

**Limitations:**

yes

**Quality:**

3

**Strengths And Weaknesses:**

## Strength
The data collection effort could be valuable to the RL community. It contributes to the endeavor of upscaling verifiable rewards

## Weakness
This work is a solid engineering effort that scales verification-based RL across broader question types. But its core contributions are largely in data collation, not in new RL algorithms or understanding model reasoning. This work definitely going to make a solid dataset track paper if it were submitted to that track.
They don’t isolate the RL fine-tuning from the dataset or verifier. Without showing how much of the performance is from increased data or from RL dynamics, it's hard to credit RL for true reasoning gains.
Among all baselines, the main complaint is that there is no standalone “Distill” comparison: Across all reported results (e.g., Tables 6–8), the paper only evaluates un fine-tuned “Base” models, supervised-fine-tuned “Instruct” models, and zero-RL variants (SimpleRL, General-Reasoner). There is no pure knowledge-distillation / direct SFT (and other no-RL) baseline to isolate the effect of the RL procedure itself.

---

> ### Author Rebuttal · Authors · 2025-07-31
>
> We sincerely thank the reviewer for acknowledging our contribution to scaling RLVR across diverse domains. We appreciate the insightful comments and valuable feedback. Below is our point-by-point response addressing each comment raised:
>
> 1. **“They don’t isolate the RL fine-tuning from the dataset or verifier.”**
>
>     In Table 4, we isolate the effect of dataset diversity by fixing the verifier and comparing performance between the full training dataset and a math-only dataset. The results clearly demonstrate that scaling training data across diverse domains improves general reasoning capabilities. Similarly, in Table 5, we isolate the effect of verifier quality by fixing the dataset and comparing our model-based verifier against a rule-based approach. This comparison highlights the advantages of using a model-based verifier for handling diverse domains. Through these controlled ablation studies, we separately examine the impact of dataset scale and verifier quality on model performance.
>
> 2. **“There is no pure knowledge-distillation / direct SFT (and other no-RL) baseline to isolate the effect of the RL procedure itself.”**
>
>     Our primary goal is to evaluate the proposed Zero-RL framework, where the model learns exclusively from questions and verifiable answers without supervised CoT labels. Therefore, our main baselines include Zero-RL methods, such as SimpleRL.
>
>     We compared our approach against models like Qwen2.5-14B-Instruct, which were trained with supervised fine-tuning (SFT). Our model consistently outperforms these models, highlighting the benefits of our RL-based training. This is particularly significant considering our training data consists of questions with verifiable answers but no explicit CoT supervision.
>
>     To further enrich our analysis, we conducted additional experiments:
>
>     - Training models directly with SFT to generate answers without reasoning steps.
>     - Training models with the original WebInstruct dataset (without verifiable answer extracted), which provides a relatively direct comparison between CoT-SFT and Zero-RL approaches.
>
>     The following table presents additional experimental results for Qwen3-4B-Base and Qwen3-14B-Base models:
>
>     | Model | Post Training | MMLU-PRO | SuperGPQA |
>     | --- | --- | --- | --- |
>     | Qwen3-4B-Base | - | 0.516 | 0.254 |
>     | qwen3-4b-webinstruct-verified-sft | SFT | 0.4397 | 0.281 |
>     | qwen3-4b-webinstruct-sft | SFT | 0.4386 | 0.227 |
>     | General-Reasoner-Qw3-4B | RL | 0.628 | 0.325 |
>     |  |  |  |  |
>     | Qw3-14B-Base | - | 0.642 | 0.365 |
>     | qwen3-14b-webinstruct-verified-sft | SFT | 0.5332 | 0.323 |
>     | qwen3-14b-webinstruct-sft | SFT | 0.4465 | 0.2542 |
>     | General-Reasoner-Qw3-14B | RL | 0.703 | 0.399 |
>
>     These results show that SFT on strong base models like Qwen3-4B-Base and Qwen3-14B-Base may even have a negative impact, while RL over our curated dataset consistently improves performance compared to the base model.
>
> 3. **“Did the authors compare the verifier against Gemini generated answers?”**
>
>     Yes, our verifiers were evaluated against Gemini-generated answers. We will add more detailed information about this comparison in Section 5.3 of the manuscript.
>
> 4. **“How is generative verifier v.s. scalar RLVR.”**
>
>     To clarify the effectiveness of our generative verifier compared to scalar verifier, we trained a scalar verifier with the same backbone and training data and evaluated its **agreement with the Gemini-based verifier** (matching Figure 5). The scalar verifier judges answers based on the probability of Pr(True).
>
>     | Answer Type | rule_verifier | scalar_verifier | model_verifier |
>     | --- | --- | --- | --- |
>     | Boolean | 0.233 | 0.851 | 0.882 |
>     | Expression | 0.192 | 0.542 | 0.743 |
>     | Float | 0.105 | 0.477 | 0.755 |
>     | Fraction | 0.699 | 0.839 | 0.924 |
>     | Integer | 0.460 | 0.680 | 0.829 |
>     | List | 0.065 | 0.472 | 0.736 |
>     | Matrix | 0.515 | 0.669 | 0.840 |
>     | Multiple Choice | 0.336 | 0.761 | 0.869 |
>     | Other | 0.106 | 0.365 | 0.681 |
>     | Percentage | 0.277 | 0.502 | 0.683 |
>     | String | 0.187 | 0.668 | 0.826 |
>
>     These results demonstrate that our generative verifier consistently outperforms the scalar verifier across all answer types. This performance advantage can come from the generative verifier's ability to leverage chain-of-thought reasoning during the verification process.
>
>     In addition to automatic evaluation comparing against the Gemini verifier, we also conducted human evaluation on 200 randomly sampled question-answer pairs to assess the **agreement with human judgment** as presented below:
>
>     | Answer Type | rule_verifier | gemini_verifier | scalar_verifier | general-verifier (ours) |
>     | --- | --- | --- | --- | --- |
>     | Boolean | 0.750 | 1.000 | 1.000 | 1.000 |
>     | Expression | 0.516 | 0.935 | 0.806 | 0.806 |
>     | Float | 0.585 | 0.892 | 0.754 | 0.877 |
>     | Fraction | 0.750 | 1.000 | 1.000 | 1.000 |
>     | Integer | 0.852 | 0.963 | 0.926 | 0.963 |
>     | List | 0.556 | 0.778 | 0.778 | 0.889 |
>     | Matrix | 0.667 | 1.000 | 0.833 | 0.833 |
>     | Multiple Choice | 0.773 | 0.864 | 0.864 | 0.909 |
>     | Other | 0.000 | 1.000 | 0.500 | 1.000 |
>     | Percentage | 0.600 | 1.000 | 0.600 | 0.700 |
>     | String | 0.312 | 0.938 | 0.812 | 0.812 |
>
>     These results further confirm that our general verifier is more effective than the scalar verifier.
>
> 5. **“Alternative verifiers? What if the verifier is trained only with open LLMs? “**
>
>     Our main contribution is scaling Zero-RL training across general domains, with the verifier being a secondary contribution. Future work could explore alternative training strategies or using open-source LLMs for verification, as well as testing various combinations of verifier models. We will acknowledge this limitation and outline these future directions in our manuscript.
>
>
> We hope our responses have addressed the reviewer's main questions and concerns. We welcome further discussion during the reviewer-author discussion period.

---

### Official Review · Reviewer_87sm · 2025-07-03

**Clarity:** 4
**Significance:** 3
**Originality:** 3
**Rating:** 5
**Confidence:** 4

**Summary:**

This paper proposes General-Reasoner, a new Zero-RL training paradigm for improving the reasoning capabilities of LLMs across diverse domains. To achieve this, a large-scale, diverse-domain dataset with high-quality reasoning questions from web resources is constructed. Additionally, a generative verifier model is trained for evaluating answers, providing robust and reliable reward signals for RL training.

**Questions:**

n/a

**Ethical Concerns:**

["NO or VERY MINOR ethics concerns only"]

**Final Justification:**

I have read the authors' response and agree that the limitation in human-based quality control should be discussed in the paper.

I will retain my score to reflect my support for this work.

**Limitations:**

yes

**Quality:**

4

**Strengths And Weaknesses:**

**Strengths**

-	This paper tackles an important gap in LLM reasoning that most prior work has primarily focused on mathematical and coding domains due to the abundance of data and the ease of answer verification.
-	The curated dataset, comprising 230K high-quality reasoning questions across various domains (e.g., physics, chemistry, finance), is a significant contribution and will likely benefit future research in general-domain reasoning.
-	Experiments are comprehensive, involving a wide range models, such as Qwen2.5 and Qwen3 family, RL models, reasoning models, and multiple evaluation benchmarks (e.g., MMLU-Pro, SuperGPQA). General-Reasoner outperforms baselines on most datasets. Additionally, ablation studies effectively analyze the impact of data abundance and model-based verifier.
-	The paper is well-written and easy to follow.

**Weaknesses**

-	The technical novelty is relative limited, but the main contributions lie in the curated dataset, the new Zero-RL training paradigm with a compact model-based verifier, and thorough experiments.
-	The data collection process heavily relies on Gemini models. Specifically, Gemini-1.5-Pro is used to extract questions, Gemini-2.0-Flash is applied to annotate each question, and solutions are generated by Gemini-2.0-flash. The potential issues introduced by these models, such as factual errors or biases, are not addressed. There is a lack of quality control in the collected data. It would benefit from including human evaluation, even on a small subset, to assess data quality.

---

> ### Author Rebuttal · Authors · 2025-07-31
>
> We sincerely thank the reviewer for the valuable and encouraging feedback acknowledging our contribution in tackling an important gap in LLM reasoning. Below is our point-by-point response to the comments:
>
> 1. **“The technical novelty is relative limited, but the main contributions lie in the curated dataset, training paradigm and experiments.” (W1)**
>
>     We agree with the reviewer that our primary contributions emphasize providing a comprehensive and scalable framework rather than technical novelty. Specifically, we contribute a diverse, high-quality dataset for RLVR training across multiple domains, alongside a cohesive training paradigm that integrates "verifiable data" with a "small model based verifier." This integration enhances scalability across various data sizes and domains while simplifying the overall approach.
>
> 2. **“There is a lack of quality control in the collected data.” (W2)**
>
>     We acknowledge that the absence of human-based quality control is a limitation in our dataset curation process. Our approach relies on automatic filtering, including dropping questions that Gemini models consistently fail to solve across multiple (8) attempts. This filtering strategy helps ensure that questions in the final dataset are solvable and free from errors. We will add a discussion about implementing further human quality control in the limitations section.

---

> > ### Comment · Reviewer_87sm · 2025-08-05
> >
> > Thank you for your response. I agree that it would be helpful to discuss the limitations of human-based quality control in the paper.
> >
> > I will retain my score to reflect my support for this work.

---

> > > ### Author Response · Authors · 2025-08-05
> > >
> > > Thank you for your thoughtful review and suggestions. We will make sure to incorporate your feedback into the revised version of our manuscript.

---

### Official Review · Reviewer_nUFT · 2025-07-03

**Clarity:** 2
**Significance:** 3
**Originality:** 3
**Rating:** 4
**Confidence:** 3

**Summary:**

This paper proposes **General-Reasoner**, a reinforcement learning framework for improving the general reasoning abilities of language models across diverse domains—ranging from math and science to social science and finance. Unlike standard RLHF pipelines, the method performs reinforcement learning **without any supervised fine-tuning**, a setting the authors call “Zero-RL.”

The paper introduces two main components:

1. **WebInstruct-Verified Dataset**: A large-scale, high-quality, and verifiable multi-domain QA dataset collected and filtered using LLMs (e.g., Gemini) to ensure answerability and consistency.
2. **Generative Verifier**: A lightweight, CoT-based model trained to semantically evaluate the correctness of answers. It serves as a reward model during RL training via GRPO (Group Relative Policy Optimization).

The method is tested on 12 benchmarks, including MMLU-Pro, GPQA, SuperGPQA, and various math datasets. It achieves state-of-the-art or highly competitive performance across the board—even with smaller models (e.g., Qwen3-4B)—without using any instruction-tuning.

**Questions:**

1. Why does Zero-RL work so well at 4B scale in your setting? Could you provide analysis (e.g., with/without verifier, or smaller RL steps) to explain?

2. Have you tested whether the model is overfitting to the generative verifier's patterns? For example, can it fool the verifier with shallow patterns?

3. How robust is the generative verifier to answer formatting, over-precision (e.g., significant figures), or semantic ambiguity?

4. How would you position this work in relation to DeepSeek Zero or other recent Zero-RL efforts?

**Ethical Concerns:**

["NO or VERY MINOR ethics concerns only"]

**Final Justification:**

I carefully read the rebuttal, other reviewers' comments, and the manuscript once again. I still found that the technical convergence of the paper to prior art is limited (the rebuttal on these points is not satisfactory), and needs a clearer explanation on why the paper can increase general reasoning ability, with a precise distinction from prior works. However, I also agree that the experimental results presented in the paper are worth noting for the community, and thus increase the score from 3 to 4.

**Limitations:**

No limitation section

**Quality:**

3

**Strengths And Weaknesses:**

## Strengths

- **Conceptual clarity and practical significance**: The method aims to train a general-purpose reasoning agent without SFT, a meaningful contribution given the costs and biases introduced by supervised pipelines.

- **Innovative dataset and reward design**: The use of LLM-filtered, verifiable multi-domain questions and a generative verifier offers a new approach to constructing diverse and semantically grounded reward signals.

- **Strong empirical performance**: General-Reasoner consistently outperforms base models and strong RL baselines across diverse tasks and domains, even when using small-scale models like 4B.

- **Readable and well-structured**: The paper is clearly written, with logically organized sections, clear diagrams, and helpful qualitative examples.

---

## Weaknesses

- **Related work on Zero-RL is not acknowledged**: Recent work such as **DeepSeek Zero** also demonstrates RL training without SFT. While General-Reasoner offers distinct innovations (e.g., the generative verifier and diverse domain coverage), the paper does not acknowledge or contrast itself with these methods, leaving its novelty under-specified.

- **Unclear why Zero-RL works at small scale**: The paper shows that 4B models trained with Zero-RL achieve strong results, but does not investigate *why*. Are the gains due to data quality, the verifier, the GRPO algorithm, or something else? No ablations or training curve analysis are provided to support this important finding.

- **Potential risk of reward hacking is not addressed**: Model-based reward signals are known to be vulnerable to “hacking” (i.e., models learning to game the verifier without improving actual reasoning), as highlighted in recent work (e.g., arXiv:2505.22203). The paper does not evaluate whether this risk affects its results, nor does it include human or external evaluation to verify reasoning quality.

- **Verifier correctness is assumed but not deeply evaluated**: The paper claims that the generative verifier is more flexible than rule-based ones, which is likely true. However, it does not assess whether the verifier enforces domain-specific norms—e.g., in a chemistry case where it marks “26.32” correct instead of “26.3,” violating significant figure rules. This could lead to inaccurate training feedback.

- **WebInstruct-Verified dataset lacks transparency**: While the data curation pipeline uses multi-answer voting and LLM filters, the thresholds for acceptance, coverage across domains, and error rates are not reported. No human validation is used to ensure that retained samples are factually or pedagogically sound.

- **No statistical significance or variance reporting**: The paper reports average accuracies but omits standard deviations or confidence intervals. Given that generalization is known to be sensitive to seeds and sampling, this limits trust in the results.

---

> ### Author Rebuttal · Authors · 2025-07-31
>
> We sincerely thank the reviewer for acknowledging our strengths in innovative dataset creation and reward design, as well as our empirical performance. We also appreciate the insightful comments and valuable feedback. Below is our point-by-point response addressing each comment raised:
>
> 1. **“Related work on Zero-RL is not acknowledged” (W1, Q4)**
>
>     Our work builds upon recent Zero-RL methods, including DeepSeek Zero and SimpleRL, as acknowledged in the Introduction and Related Work (e.g., line 26, line 102). However, our primary contribution differs in both motivation and execution. While existing Zero-RL approaches primarily focus on mathematical reasoning, we explicitly aim to enhance general reasoning across diverse domains by significantly expanding verifiable training data and evaluations. We will further clarify this distinction in the related work section.
>
> 2. **“Unclear why Zero-RL works at small scale” (W2, Q1)**
>
>     Figure 4 presents the training curve for the Qwen3-4B model. The initial rapid gains likely stem from the model aligning its answer formats with expectations, while continued improvements increasingly depend on data quality and verifier effectiveness. We will include more detailed discussions results section.
>
> 3. **“Potential risk of reward hacking is not addressed” (W3, Q2)**
>
>     In the early stages of our experiments, we did observed risks of reward hacking. The policy model tended to inject long content into the latex box for short answers. This resulted in increasing training rewards but poor test performance.
>
>     As mentioned in line 200, our reward design thus includes penalties for answers that significantly deviate in length from the ground truth short answers. This constrains the model's ability to inject hacking content intended to mislead the verifier.
>
>     Our comprehensive evaluation across various downstream reasoning tasks demonstrates that our verifier-guided training produces robust performance improvements, suggesting that the resulting model is not affected by the potential risk.
>
> 4. **“How robust is the generative verifier to answer formatting” (W4, Q3)**
>
>     In Figure 5, we evaluate the agreement between our smaller generative verifier and the Gemini-based verifier, demonstrating high alignment between the two.
>
>     However, we acknowledge the reviewer's concern regarding the assumed correctness of the verifier.
>
>     Regarding the reviewer's example of a chemistry case where "26.32" is marked correct instead of "26.3," this illustrates precisely why we need a model-based verifier that makes judgments conditioned on question context.
>
>     To assess verifier quality, we manually evaluated 200 randomly sampled question-answer pairs, comparing the verification agreement of different verifiers with human verification. Below are the results:
>
>     | Answer Type | rule_verifier | gemini_verifier | general-verifier (ours) |
>     | --- | --- | --- | --- |
>     | Boolean | 0.750 | 1.000 | 1.000 |
>     | Expression | 0.516 | 0.935 | 0.806 |
>     | Float | 0.585 | 0.892 | 0.877 |
>     | Fraction | 0.750 | 1.000 | 1.000 |
>     | Integer | 0.852 | 0.963 | 0.963 |
>     | List | 0.556 | 0.778 | 0.889 |
>     | Matrix | 0.667 | 1.000 | 0.833 |
>     | Multiple Choice | 0.773 | 0.864 | 0.909 |
>     | Other | 0.000 | 1.000 | 1.000 |
>     | Percentage | 0.600 | 1.000 | 0.700 |
>     | String | 0.312 | 0.938 | 0.812 |
>
>     These results demonstrate the high accuracy and robustness of our generative verifier across diverse answer formats. This human evaluation further confirms the significant advantage of our generative verifier compared to traditional rule-based methods. We will incorporate this detailed analysis based on human evaluation into the revised manuscript.
>
> 5. **“No statistical significance or variance reporting” (W6)**
>
>     Our evaluation protocol adheres to established practices in prior research. For high-variance datasets like AIME24/25, we report performance averaged over 32 runs, consistent with methods such as Simple-RL. For comprehensive benchmarks like MMLU-Pro and SuperGPQA, we report averages across large test sets containing more than 10,000 queries each, following the protocols established by the original datasets.
>
> 6. **“WebInstruct-Verified dataset lacks transparency” (W5)**
>
>     We would like to highlight that the WebInstruct-Verified dataset will be publicly released. To further improve transparency, we will also provide the unfiltered dataset and detailed outputs from the Gemini model used in data curation. This additional information will ensure complete transparency in our dataset creation process.
>
>
> We hope our response has addressed your main questions and concerns. Your suggestions have helped us improve the paper. We welcome further discussion during the reviewer-author discussion period.

---

> > ### Author Response · Authors · 2025-08-07
> >
> > We sincerely thank you for taking the time to carefully review our manuscript. As there is only one day remaining in the reviewer–author discussion period, we would greatly appreciate it if you could let us know whether our rebuttal has sufficiently addressed your major concerns.
> >
> > If there are any remaining issues or additional comments regarding our responses, we would be more than happy to clarify or provide further explanation.

---

> > ### Comment · Reviewer_nUFT · 2025-08-07
> > **Reply**
> >
> > Thank you for providing a rebuttal. After I carefully read the rebuttal, other reviewers' comments, and the paper once again, I still think the clarity (especially technical convergence to prior art) needs to be significantly improved. For example, the author replies that a penalty on answer length helps prevent the reward hacking problem. Is it generally known, or novel findings? Since reward hacking is a classic problem of RL, there has been extensive research on the front. While this is just an example, and such a lack of careful explanation about the relationship with prior art makes me feel the paper less scientific.
> >
> > However, at the same time, I also acknowledge that the paper provides extensive and strong empirical evidence, and it is worth noting for the community. Given that, I slightly increased my score, I want to leave the final decision to AC.

---

> > > ### Author Response · Authors · 2025-08-08
> > >
> > > We sincerely thank you for highlighting our strong empirical evidence and contributions to the community, as well as for your willingness to raise the score. We greatly appreciate your insightful reviews and constructive discussions. At the same time, we value the comments regarding the clarity of our discussion in relation to prior work. We will ensure that your suggestions are thoroughly addressed in the updated manuscript.

---

> ### Author Response · Authors · 2025-08-05
>
> We thank the reviewer for carefully reviewing our manuscript. Given that only a few days remain in the reviewer–author discussion period, we kindly ask if our rebuttal has addressed your initial major concerns. If there are any additional comments regarding our rebuttal. We would be very happy to further address them.

---

### Author Response · Authors · 2025-08-09
**End of Discussion Period Notes**

We sincerely thank all reviewers for their insightful feedback and constructive discussions. We will ensure that all suggestions are thoroughly addressed in the revised manuscript.

---

### Note · Authors · 2025-08-12

We thank all reviewers for their thoughtful feedback and suggestions. We appreciate their acknowledgment of our contribution of the proposed training framework: scaling RLVR training beyond math through a diverse, verifiable training data and a compact generative verifier, and for considering score increases or maintaining positive assessments after rebuttal. Below, we summarize the key issues addressed in our rebuttal.

**Summary of Acknowledged Strengths**

- Tackles general-domain reasoning (beyond math) with consistent gains on MMLU-Pro, GPQA/SuperGPQA, and BBEH.
- Contribution of a 230K multi-domain, verifiable RLVR training dataset.
- Use of a lightweight generative verifier that handles various answer formats to make the training effective.
- Thorough experimental evaluation.

**Summary of Concerns Addressed During Rebuttal**

1. Novelty compared to existing/concurrent work.
    - We clarified our scalable training framework for general-domain RLVR: (i) diverse, verifiable QA data and (ii) a compact generative verifier. This approach differs from math-focused Zero-RL/GRPO (e.g., DeepSeek, SimpleRL) and from the original WebInstruct SFT.
2. Additional experiments requested by reviewers:
    - SFT baselines: SFT on WebInstruct-Verified and WebInstruct significantly underperforms our approach with a substantial performance gap.
    - Human evaluation of verifier results: Our generative verifier demonstrates the highest agreement with human judgments based on 200 randomly sampled questions, comparing to scalar-based verifier and rule-based verifier.
    - Extended training: Our training demonstrates consistent, non-spurious improvements.
3. Additional discussions:
    - Clearer positioning relative to concurrent work.
    - Dataset transparency (releasing intermediate datasets from the processing pipeline).
    - Reward-hacking mitigation (discussion on length-deviation penalties).
4. Expanding the limitation discussion:
    - Scope limited to short-answer RLVR.
    - Reliance on Gemini during the curation process.
    - Further discussion of reward-hacking concerns and general-verifier robustness.

We will incorporate these updates in the final version and release code, data, checkpoints, and all curation/verifier artifacts. We hope our work paves the way for enhancing LLM reasoning capabilities across broad domains and tasks. Thank you again for the constructive guidance.

---

### Decision · Program_Chairs · 2025-09-17

**Decision:**

Accept (poster)

**Comment:**

This paper presents a pipeline for constructing a large-scale reasoning dataset from web-crawled data. Using this pipeline, the authors curate 230K high-quality reasoning questions spanning a wide range of reasoning domains. The dataset is then used to train a domain-agnostic model-based verifier for evaluating answers. Within a Zero-RL training framework, the dataset and verifier yield reasoning models with improved accuracy across multiple domains.

While the method itself is not highly novel, the data curation pipeline is sound, the paper is well written, and the empirical results are comprehensive. The proposed dataset and verifier should prove valuable for researchers and practitioners working on reasoning problems.

Given the quality of the dataset, the strength of the results, and the potential impact on the field, AC recommends acceptance.